# Endosomal v-ATPase as a Sensor Determining Myocardial Substrate Preference

**DOI:** 10.3390/metabo12070579

**Published:** 2022-06-22

**Authors:** Shujin Wang, Yinying Han, Miranda Nabben, Dietbert Neumann, Joost J. F. P. Luiken, Jan F. C. Glatz

**Affiliations:** 1Institute of Life Sciences, Chongqing Medical University, Chongqing 400032, China; shujin.wang@cqmu.edu.cn (S.W.); hanyinying@stu.cqmu.edu.cn (Y.H.); 2Department of Genetics & Cell Biology, Faculty of Health, Medicine and Life Sciences, Maastricht University, 6200 MD Maastricht, The Netherlands; m.nabben@maastrichtuniversity.nl (M.N.); j.luiken@maastrichtuniversity.nl (J.J.F.P.L.); 3CARIM School for Cardiovascular Diseases, 6211 LK Maastricht, The Netherlands; d.neumann@maastrichtuniversity.nl; 4Maastricht University Medical Center, Department of Clinical Genetics, 6229 HX Maastricht, The Netherlands

**Keywords:** heart, vacuolar-type H^+^-ATPase, endosomal acidification, lipid, glucose, amino acids, CD36, GLUT4

## Abstract

The heart is a metabolically flexible omnivore that can utilize a variety of substrates for energy provision. To fulfill cardiac energy requirements, the healthy adult heart mainly uses long-chain fatty acids and glucose in a balanced manner, but when exposed to physiological or pathological stimuli, it can switch its substrate preference to alternative substrates such as amino acids (AAs) and ketone bodies. Using the failing heart as an example, upon stress, the fatty acid/glucose substrate balance is upset, resulting in an over-reliance on either fatty acids or glucose. A chronic fuel shift towards a single type of substrate is linked with cardiac dysfunction. Re-balancing myocardial substrate preference is suggested as an effective strategy to rescue the failing heart. In the last decade, we revealed that vacuolar-type H^+^-ATPase (v-ATPase) functions as a key regulator of myocardial substrate preference and, therefore, as a novel potential treatment approach for the failing heart. Fatty acids, glucose, and AAs selectively influence the assembly state of v-ATPase resulting in modulation of its proton-pumping activity. In this review, we summarize these novel insights on v-ATPase as an integrator of nutritional information. We also describe its exploitation as a therapeutic target with focus on supplementation of AA as a nutraceutical approach to fight lipid-induced insulin resistance and contractile dysfunction of the heart.

## 1. Introduction

An optimal contractile function of the heart is required to maintain a sufficient supply of oxygen and nutrients to peripheral tissues [1,2,3]. To maintain this contractile function the heart needs to receive a continuous supply of substrates to produce energy, in the form of energy-rich phosphate bonds (i.e., adenosine triphosphate, ATP) [4]. Most of this ATP is derived from the mitochondrial oxidation of various energy substrates, in particular (long-chain) fatty acids and carbohydrates (e.g., glucose and lactate), but also ketone bodies and amino acids (AAs) [2,5]. These latter substrates are commonly referred to as “alternative substrates”. At present, it is generally accepted that fatty acids and glucose compete as a major source of acetyl-CoA for entering the tricarboxylic acid cycle and subsequent mitochondrial oxidative energy provision [6]. Healthy adult individuals predominantly utilize fatty acids for cardiac energy generation (through β-oxidation, contribution ~60%), while glucose and lactate are responsible for ~30% of the total energy production (glycolysis and glucose oxidation) [7,8] (Figure 1). Additionally, the omnivoric character of the heart implies that, depending on physiological conditions, the alternative substrate’s ketone bodies and AAs are also being used for cardiac energy provision [2,6,9]. In the healthy heart, the contribution of these alternative substrates to total ATP production is limited, with ketone bodies and AAs accounting for 10% and 2%, at most, of total ATP generated, respectively [6]. However, at any specific time the actual contribution of each of these substrates could be quite different and can be adjusted rapidly, depending on changes in substrate availability governed by specific (patho)physiologic conditions (e.g., changes during the course of a day) [2,10,11,12].

The heart in physiological conditions is flexible to rapidly and reversibly shift between the source of substrates, especially with respect to fatty acids and glucose, depending on the supply. However, a chronic shift towards a predominant use of fatty acids or glucose often leads to cardiac dysfunction [2]. In the case of the preferred utilization of one substrate over another, there is a risk that the heart suffers from fuel toxicity, i.e., lipo- or glucotoxicity, each being a condition that elicits major impairments of cardiac functioning [13,14] (Figure 1). Thus, re-balancing cellular fuel supply, in particular with respect to fatty acids and glucose, may be an effective strategy to treat the failing heart [2,15].

There is increasing evidence that vacuolar-type H^+^-ATPase (v-ATPase) is a novel key regulator of myocardial substrate preference. v-ATPase is the main protein complex involved in the maintenance and regulation of the pH of intracellular organelles, i.e., endosomes and lysosomes, but also serves other roles connected to its proton pumping activity. Notably, v-ATPase is involved in the regulation of the cellular uptake of lipids, glucose, and AAs. Vice versa, the pump function of v-ATPase is under the regulation of these substrates, disclosing it as a novel energy sensor. In this way, v-ATPase is involved in several metabolic feedback loops. Finally, v-ATPase may integrate all this nutritional information, making it an attractive target for therapeutic strategies to treat heart diseases with metabolic roots.

In the present review, we first highlight v-ATPase’s structure and function (Section 2), then we reveal the crosstalk between v-ATPase and lipids (Section 3), after which we will disclose the reciprocal regulation of v-ATPase activity with glucose (Section 4) and with amino acids (Section 5). Finally, based on these insights, we discuss possible strategies to combat heart diseases via the regulation of v-ATPase function.

## 2. Structure and Function of v-ATPase

Structurally, v-ATPase is composed of at least 14 different subunits that are organized into two sub-complexes: a membrane-associated sub-complex (named V_0_) of 6 subunits forming the proton translocation channel and a cytoplasmic sub-complex (V_1_) of 8 subunits forming the ATP-driven rotor [16,17,18,19,20,21,22]. This subunit composition appears highly conserved from yeast to mammals [23]. The membrane-embedded V_0_ domain, where proton translocation occurs, includes subunits a, c, c″, d, e, and the accessory subunits Ac45 and M8-9 [24,25,26]. The a-subunit possesses two domains, an N-terminal portion that interacts with the V_1_ subunit and a C-terminal portion that is embedded within the membrane and participates in proton translocation. The V_0_ and V_1_ domains are connected by a central stalk composed of the D and F subunits and by three peripheral stalks formed by the C, E, G, and H subunits, as shown in (Figure 2), [21,24,27,28].

v-ATPase is present in acidic intracellular membrane compartments such as lysosomes, endosomes, and secretory vesicles. In specific mammalian tissues, v-ATPase is also partly localized to the plasma membrane [29,30]. The proton pumping into lysosomes, endosomes, and secretory vesicles influences many processes associated with these organelles, including vesicular trafficking, endocytosis, autophagy, receptor recycling, and protein degradation [19,24]. The v-ATPase-dependent acidification of early endosomes provides the low pH signal that causes endocytosed ligands, such as low-density lipoproteins, to dissociate from their receptors [29,31]. In turn, this dissociation is required for recycling the receptors to the plasma membrane and targeting the released ligands to the lysosome for degradation. Endosomal acidification is also involved in the budding of endosomal carrier vesicles that transport cargo between early and late endosomes [32] and in the trafficking of newly synthesized lysosomal enzymes from the Golgi to the lysosome utilizing the mannose-6-phosphate (M6P) receptor, the latter of which interacts with lysosomal enzymes bearing a M6P recognition marker in a pH-dependent manner [33]. Recently, v-ATPase has been shown to function in the earliest stages of clathrin-coated vesicle formation [34]. From studies in yeast, *Drosophila melanogaster*, *Caenorhabditis elegans*, and *Mus musculus* there is also evidence that the integral V_0_ domain of the v-ATPase may play a role in membrane fusion independent of acidification [29,35,36,37,38,39]. Additionally, plasma membrane V_1_ domain v-ATPases are primarily present in specialized cells [29]. In osteoclasts, v-ATPases are targeted to the ruffled border in contact with bone and provide the acidic extracellular environment that is essential for bone resorption [40]. Defects in the plasma membrane v-ATPase in osteoclasts lead to loss of bone resorption and the development of the disease osteopetrosis, which is characterized by highly brittle bone and skeletal defects during embryonic development [40]. In renal alpha intercalated cells of the late distal tubule and collecting duct, v-ATPase is targeted to the apical membrane and is involved in acid secretion into the urine [26]. The density of v-ATPase complexes in the apical membrane of intercalated cells is tightly controlled in response to plasma pH, which occurs through both exocytic insertion and endocytic retrieval of pumps. A decrease in plasma pH results in an increase in the number of v-ATPases at the apical surface which, in turn, increases acid secretion into the urine. Therefore, v-ATPase is important for pH homeostasis in kidney [29]. In the male reproductive tract, v-ATPase is present in the apical membrane of epididymal clear cells and functions to maintain an acidic pH in the epididymal lumen that is essential for normal sperm development and storage [26]. Finally, v-ATPase has been implicated in left–right patterning during vertebrate development, with the pump’s ability to generate a membrane potential apparently key to its function [41,42].

Studies in yeast and mammalian kidney cells revealed that v-ATPase activity is mainly regulated via assembly and disassembly of the V_1_/V_0_ complex [22,25,43,44]. Hence, V_1_/V_0_ assembly/disassembly appears highly conserved in eukaryotes [23]. Moreover, V_1_/V_0_ assembly/disassembly is rapid and reversible, so that it leads to immediate modulation of v-ATPase activity. During disassembly, V_1_ subunit C, which acts as a bridging subunit between the V_1_ and V_0_ domains, leaves the v-ATPase complex [45,46], allowing the remaining V_1_ domain to separate from the V_0_ domain. A second V_1_ subunit, subunit H, then undergoes a conformational change that prevents free V_1_ from hydrolyzing ATP, preventing energy depletion in the absence of proton transport [47,48,49,50]. Upon dissociation, passive proton translocation across free V_0_ is blocked [51]. Therefore, disassembly inhibits v-ATPase function in vivo. This process is fully reversible upon the re-addition of V_1_ subunit C [45]. In insect cells, re-assembly requires the phosphorylation of subunit C [52], although it is unclear if this also happens in fungi and humans, and if so, which kinase(s) is involved. Notably, although disassembly involves microtubules [53], no other known disassembly factors are involved. Additionally, catalytically active v-ATPase is required for disassembly in both yeast and human cells [47,54,55]. Re-assembly uses a v-ATPase-exclusive chaperone [56]. These latter findings suggest that v-ATPase is naturally prone to disassemble [57].

The number of stimuli identified to alter v-ATPase assembly is increasing [17,25,58]. While in heart cells, glucose was already known for two decades to induce v-ATPase disassembly, lipids (e.g., palmitate and oleate) and AAs (e.g., arginine, leucine, and lysine) can be added to the list of metabolites/nutrients regulating v-ATPase function [59,60,61]. Together, these recent findings suggest that in the heart v-ATPase senses and integrates nutritional information (e.g., palmitate, glucose, and AAs), as discussed in more detail in subsequent sections.

## 3. Crosstalk between Lipids and v-ATPase

### 3.1. Lipid and CD36

Lipid (or fatty acid) uptake into the heart is regulated by several membrane-associated fatty acid transporters, including fatty acid-binding protein at the plasma membrane (FABP_PM_), fatty acid-transport proteins (FATP) 1–6, caveolin-1, and fatty acid translocase/cluster of differentiation 36 (FAT/CD36, officially known as scavenger receptor-B2) [62,63]. From these fatty acid transporters, CD36 is the major contributor to adaptive fatty acid uptake into the heart [62]. CD36, a 472-amino acid (88 kDa) membrane protein, is not only localized to the sarcolemma, but also present in endosomal membranes [63,64]. Under basal conditions in skeletal muscle [65] and heart [66], an estimated 50% of CD36 is stored in endosomes. Endosomes are acidic organelles with a luminal pH between 5.5 and 6.0. This luminal acidity is important for CD36 storage [67]. In response to physiological stimuli, such as insulin or contraction, CD36 translocates to the sarcolemma. Specifically, CD36 translocation occurs via vesicle-mediated trafficking, which includes vesicle budding from endosomes, translocation along cytoskeletal filaments, and vesicle fusion with the sarcolemma [63,68]. Subsequently, the sarcolemmal localization of CD36 then enables an increase of fatty acid uptake, e.g., in order to immediately meet the changing energy demand, e.g., for muscular contractions [62]. The actual rate of cellular fatty acid uptake is determined by the fatty acid gradient across the sarcolemma, with low intracellular fatty acid concentrations due to rapid enzymatic conversion to fatty acyl-CoA by long-chain acyl-CoA synthetase (ACSL) [67].

What happens with myocellular fatty acid uptake during lipid overconsumption? When the fatty acid concentration in the circulation rises, such as upon increased lipid consumption, the rate of fatty acid uptake increases, as governed by an increased transmembrane gradient of fatty acids and enabled by the facilitatory action of CD36. In line with this, cardiomyocytes pretreated with CD36-blocking antibodies [69] as well as hearts from CD36 knockout mice are protected from lipid overload in vitro and in vivo, respectively [70,71]. Subsequently, increased fatty acid flux through CD36 is followed by increased CD36 translocation from endosomes to the cell surface, thereby further increasing fatty acid uptake. Hence, during lipid overload, there is an installment of a vicious cycle of CD36-mediated CD36 translocation. In conclusion, these findings demonstrate CD36 to be a key player in increased fatty acid uptake in lipid overload conditions [2,67,72].

The mechanism by which lipids (e.g., palmitate) increase CD36 translocation to the sarcolemma is incompletely understood, but recently it became evident that excess lipids inhibit proper endosomal acidification, which is dependent on v-ATPase [59]. The involvement of v-ATPase in CD36 translocation and the consequences of increased CD36-mediated fatty acid uptake on v-ATPase activity will be explained in the next section.

### 3.2. v-ATPase and CD36 Traffic

Proper v-ATPase activity is essential for endosomal acidification. In turn, acidification of the endosomes is necessary for these organelles to serve as a storage compartment for CD36. Hence, endosomal retention of CD36 is dependent on the proper functioning of v-ATPase [59,60,61]. Conversely, interfering with endosomal acidification, either by treatment of cardiac cells with monensin, a proton ionophore, or with bafilomycin-A, a v-ATPase inhibitor, causes expulsion of CD36 from the endosomes and subsequent translocation to the cell surface [73]. Additional evidence for the involvement of v-ATPase in intracellular retention of CD36 comes from genetic manipulation of v-ATPase via siRNA-mediated silencing of the V_1_B2 subunit. This silencing also causes the depletion of other subunits, and hence, reduces the formation of the v-ATPase holo-complex. As a result, there is a loss of acidification of intracellular compartments, which is then followed by increased CD36 translocation and subsequent excessive myocellular lipid accumulation [59]. We currently do not know why endosomal acidification is important for CD36 retention, but it is speculated that a low endosomal pH keeps the primary amine groups in phospholipids (e.g., phosphatidylethanolamine) in a protonated form, which favors stabilization of the bilayer (because of ion-pairing with negatively charged phosphate groups of adjacent phospholipids) [67]. Such bilayer stabilization would impair membrane curving events [74], so that the budding of (CD36-containing) transport vesicles would become more difficult [67].

What does the reliance on proper endosomal acidification imply for the functioning of the endosomes in serving as CD36 storage compartments in cardiac cells during lipid overload? Excess lipid taken up into cardiac cells, e.g., HL-1 cells, adult rat cardiomyocytes (aRCMs), or human induced pluripotent stem cell-derived cardiomyocytes (hiPSC-CMs) via CD36, was observed to lead to impairment of v-ATPase function and loss of endosomal acidification [59,60,61]. The mechanism behind this lipid-induced v-ATPase inhibition appears to be the disassembly of V_1_ from V_0_. Supporting evidence for this mechanism were the observations that overexposure of cardiac cells to lipids (palmitate) (i) decreased the co-immunoprecipitation between two subunits on different sub-complexes (V_0_a2 and V_1_B2), and (ii) decreased the association of subunit V_1_B2 (within the cytoplasmic V_1_ sub-complex) with endosomal membranes [59,60]. Furthermore, microscopic studies show extensive co-localization of CD36 with V_0_a2 and V_1_B2 in cardiomyocytes cultured under low palmitate conditions [75].

At present, we can only speculate how lipid oversupply may cause v-ATPase disassembly. There are arguments for at least two possibilities: (i) In neuronal cells v-ATPase disassembly appears to be induced by palmitoylation [76]. As various proteins undergo palmitoylation and oleoylation at the same cysteine residue [77], the disassembly signal might include long chain-fatty acylation of proteins involved in v-ATPase regulation, such as intracellular signaling molecules, i.e., which would change their activity or localization to induce disassembly. However, such a palmitoylation target capable of regulating v-ATPase assembly status remains elusive. (ii) Another possible explanation might include the involvement of AMP-activated protein kinase (AMPK) in the control of v-ATPase assembly. This heterotrimeric enzyme complex, consisting of a catalytic α-subunit, and regulatory γ-subunit, connected by a bridging β-subunit, plays a key role in cellular energy homeostasis. All these subunits have at least two different isoforms, and AMPK can be assembled from all combinations. Relevant to the link with v-ATPase is the observation that CoA esters of palmitate allosterically activate AMPK (but only the β1-containing heterotrimers). This AMPK activation serves to increase fatty acid oxidation by phosphorylating acetyl-CoA carboxylase [78]. Specifically, AMPK via the β1 subunit, together with the scaffolding protein axin, is recruited to lysosomally localized v-ATPase (with the help of another adaptor protein ragulator; see Section 5), where it is activated by phosphorylation of an upstream kinase (i.e., liver kinase B1, with v-ATPase being in the disassembled state [79,80]). This may then lead to v-ATPase disassembly. Given the established role of v-ATPase in AMPK activation, further studies are warranted to clarify the exact regulation of v-ATPase disassembly by fatty acyl-CoA mediated AMPK activation, specifically in the heart. In addition, it has been reported that sestrin, a metabolic regulator that responds to various stresses, including nutrition, can cause AMPK activation [81], raising the possibility of sestrin involvement in regulating v-ATPase assembly (as further discussed in Section 5.2).

The consequence of chronic cardiac lipid overload and the resulting chronically increased fatty acid uptake rate is a progressive accumulation of fatty acid metabolites, mostly triacylglycerols stored in lipid droplets. Yet, also intracellular concentrations of less inert lipid metabolites increase, such as those of diacylglycerols and ceramides [67,82]. These metabolites ultimately cause the heart to develop insulin resistance and contractile dysfunction [83]. The order of events is schematically depicted in Figure 3.

## 4. Crosstalk between Glucose and v-ATPase

### 4.1. Glucose Influx Mediates v-ATPase Function

Whereas lipids were found only recently to regulate v-ATPase assembly status, for the other main cardiac substrate, glucose, this was documented more than two decades ago [45,84]. Many proliferating cells preferentially use glucose as an energy substrate, making metabolic plasticity and adaptation to low-glucose environments critical to success [25]. The widespread use of glucose as a cellular energy substrate entails the requirement to adapt to fluctuations in glucose availability, which appears to involve v-ATPase. In particular, v-ATPase activity is increased by increases in glucose supply, which is closely associated with increased glycolysis in species ranging from the simplest single-celled eukaryotic organism to complex, multicellular mammals [25,44,45,47,60,85,86,87] (Figure 4).

Thus far observed in yeast, the best understood glucose-sensitive signaling mechanism controlling v-ATPase assembly is the Ras/cAMP/protein kinase A (PKA) pathway [88]. Active Ras is a GTP-coupled protein, and Ras activity is negatively regulated by the Ira1p and Ira2p GTPase-activating proteins (GAPs). Glucose addition inhibits Ira1p and Ira2p, and GTP-bound Ras can then activate adenylate cyclase to produce cAMP. Elevated levels of cAMP trigger dissociation of the PKA regulatory subunit to activate the kinase activity of PKA [25,88]. The downstream effect of glycolytic signaling through PKA enhances v-ATPase assembly, although the specific PKA phosphorylation target on v-ATPase that triggers complex formation is still unclear.

In mammals, the phosphoinositol-3 kinase (P13K) pathway can affect glucose-triggered assembly [44,89]. Chronic glucose depletion triggers increased v-ATPase disassembly, lowered v-ATPase activity, and defective acidification in mammalian cells. After disassembly, v-ATPase requires PI3K activity for re-assembly; specifically, re-assembly can be induced by adding back glucose or a constitutively active PI3K catalytic subunit [44]. This appears related to microfilament formation or phosphorylation of lipids to change the lipid environment [44]. The first glycolytic intermediate, glucose-6 phosphate (G6P), may be the initial signal for re-assembly in mammal cells.

In a search for potential protein partners that interact with v-ATPase, the glycolytic enzyme aldolase has been shown to bind directly to v-ATPase [90,91,92]. The v-ATPase disassembly observed in aldolase deletion mutant cells can be restored to normal levels by aldolase complementation, indicating direct coupling of glycolysis to the proton pump [90]. Accordingly, the binding of aldolase to the v-ATPase provides the cell with a means for localized ATP generation by glycolysis [91]. The interaction between aldolase and v-ATPase increased dramatically in the presence of glucose, indicating that aldolase acts as a glucose sensor for v-ATPase regulation [90] (Figure 4). This glucose-sensing mechanism of aldolase is mediated by its direct substrate: the glycolysis intermediate fructose-1,6-bisphosphate (FBP). Processing of FBP by aldolase triggers the binding of aldolase to v-ATPase [93]. Additionally, increased FBP levels disrupt the v-ATPase AXIN/LKB1 complex, thereby disallowing AMPK-mediated v-ATPase disassembly (see Section 3). Finally, FBP activates the mechanistic target of rapamycin complex 1 (mTORC1). The role of mTORC1 in v-ATPase activation is discussed in greater detail in Section 5. In a broader context, the above observations emphasize the intricate link between v-ATPase assembly status and major anabolic/catabolic cellular decision-making.

Future studies should investigate the possible roles of glycolysis in mediating v-ATPase assembly and activity, specifically in the heart.

### 4.2. v-ATPase Influences Glucose Uptake

Glucose uptake into the heart is mainly mediated by glucose transporters GLUT1 and GLUT4, whereby GLUT1 is responsible for most of the basal uptake and GLUT4 for most of the stimulus-inducible glucose uptake [2,67,94]. Concurrently, GLUT1 is mainly localized to the sarcolemma, while GLUT4 is mainly stored in intracellular compartments from where it can translocate to the sarcolemma. Similar to CD36, GLUT4 is also localized to the endosomes [95]. Hence, the subcellular GLUT4 distribution partially overlaps with that of CD36 [67]. In heart and muscle, insulin and contraction are the main physiological stimuli inducing GLUT4 translocation, which is similar to the regulation of vesicle-mediated CD36 translocation [63,67]. Hence, under normal physiological conditions, CD36 and GLUT4 recycling are similarly regulated [67,96]. In conditions of chronic lipid overload (as in type 2 diabetes), when the endosomes undergo loss of acidification, GLUT4 is expelled from the endosomes, just like CD36. However, whereas CD36 permanently translocates to the sarcolemma, GLUT4 remains intracellularly. It has been suggested that GLUT4 translocates from the endosomes to a non-endosomal intracellular compartment [71].

In contrast to CD36, which is mainly stored within the endosomes [95,97], GLUT4 appears to be stored in diverse subcellular compartments [67]. Indeed, there is strong evidence for the existence of a relatively large subcellular pool of GLUT4 outside of the endosomes, i.e., the GLUT4 storage vesicles (GSV). This pool may contain about half of the intracellularly stored GLUT4, at least in adipocytes (with the endosomes containing the other half) [67,98]. The GSV can be distinguished from the endosomes because these vesicles form tubule-vesicular structures of ~70 nm size that are non-acidified and equipped with a distinct subset of proteins, such as an insulin-regulated aminopeptidase (IRAP) [67,99,100,101]. Furthermore, GSV contains TUG (tether containing a UBX domain for GLUT4) that is responsible for GLUT4 retention within the GSV via binding to its N-terminal region [102]. In the absence of a trigger, the C-terminal region of TUG anchors GSV to the cis-Golgi and thereby imprisons GLUT4. Upon insulin stimulation, TUG undergoes an endoproteolytic cleavage which releases the GSV from the Golgi, after which these vesicles rapidly translocate to the cell surface [102,103,104]. Besides acting as a storage compartment for insulin-stimulated GLUT4 translocation, the GSV might also serve as a storage compartment for contraction-induced GLUT4 translocation [67].

Do the GSV play a role in GLUT4 traffic during lipid overload of cardiac cells? There is evidence that upon loss of endosomal acidification by treatment of cardiac cells with the v-ATPase inhibitor bafilomycin-A, GLUT4 translocates from the endosomes to the GSV [67,73]. It is plausible that GLUT4 travels in the same manner to the GSV upon loss of endosomal acidification by lipid overexposure of cardiac cells. This could be a true imprisonment for GLUT4 because during lipid overload conditions, the insulin signaling pathway is not able to induce the endoproteolytic cleavage of TUG needed for the liberation of the GSV [67]. Hence, upon full development of insulin resistance in cardiac cells, the subcellular distribution of GLUT4 becomes juxtaposed to that of CD36, with GLUT4 being inside and CD36 outside. Thus far, we still do not understand why CD36 and GLUT4 have different destinations after being expelled from the endosomes. Perhaps, differences in the protein composition of the vesicular trafficking machinery of CD36 versus GLUT4 (such as different isoforms of the protein family of vesicle-associated membrane proteins; VAMPs) may determine the default destination of both transporters when the endosomes lose their capacity to store them [67].

## 5. Crosstalk between Amino Acids and v-ATPase

### 5.1. Amino Acid Sensing Needs v-ATPase

Amino acid homeostasis, which involves balancing the production and utilization of individual amino acids, is essential to cellular and organismal viability [105,106]. To achieve this homeostasis, cells must continually sense amino acid availability and respond accordingly [105]. A major way that cells replenish the available supply of amino acids is through lysosomal proteolysis, which mainly depends upon the low luminal pH generated by v-ATPase [17,18,58,61,107].

Recently, v-ATPase was identified as a critical component of the amino acid-sensing machinery that communicates amino acid availability to the mammalian target of rapamycin complex 1 (mTORC1) [58,80,107]. Mechanistically, v-ATPase has been shown to directly associate with the Ragulator complex on the lysosomal membrane [107]. This same complex formation of v-ATPase and Ragulator expectedly also occurs on the endosomal membrane [61]. The ragulator serves as a platform for the RagGTPases which, when activated by the presence of AAs, recruit mTORC1 to lysosomes, allowing it to be activated by lysosomally localized Ras homolog enriched in brain (Rheb) [21,105,108] (Figure 5). Especially Leu and Arg appeared to be potent AAs to activate mTORC1 in a v-ATPase-dependent manner [109].

An important other aspect of v-ATPase-mediated AA sensing by mTORC1 is that AA (especially Arg) needs to be taken up into lysosomes (or endosomes) prior to mTORC1 activation [107,109]. This so-called “lysosome-centric inside-out model of AA sensing by mTORC1” apparently is an important contributor to AA-induced mTORC1 activation [107]. Another essential component of this sensing mechanism is the lysosomal AA transporter SLC38A9, which is, just like ragulator, interacting with v-ATPase. In conclusion, v-ATPase appears to operate as an essential scaffold protein involved in AA activation of mTORC1.

### 5.2. Reciprocal Regulation of v-ATPase on Amino Acid Sensing

Initial studies on the effects of AAs on v-ATPase showed that AAs inhibited rather than activated v-ATPase assembly, as found in HEK293 cells [58]. Our recent findings were in contradiction with those earlier data, in that we found AA to stimulate v-ATPase assembly in cardiac cells. We will explain the reason for this discrepancy later, but for now, we will just mention that the solution to unify these opposing observations lies in the difference in AA concentrations added to the cells. Hence, AAs appear able to activate mTORC1 and v-ATPase in a mutual manner. We confirmed that Leu and Arg were among the most potent ones, together with Lys [61]. Especially when combined into a cocktail, these three AAs were equally potent as the complete AA mixture [61]. v-ATPase activation by AAs (and also by the Arg/Leu/Lys cocktail) in cardiac cells appeared to be dependent on ragulator and on SLC38A9, as silencing of the ragulator subunit lamtor1 and of SLC38A9 annihilated this effect. This latter finding bolsters the notion of a mutual activation of mTORC1 and v-ATPase by AAs via a lysosome/endosome-centric inside-out AA sensing mechanism.

It may not be a coincidence that among the most potent AAs to induce this mutual activation mechanism, there are two basic AAs (i.e., Arg and Lys), as these AA may accumulate into endosomes/lysosomes according to the weak base trapping mechanism [61]. It has been shown that lysosomally present Arg will be sensed by SLC38A9 [109], which then may trigger a conformational change in the neighboring v-ATPase V_0_ subcomplex, thereby contributing to v-ATPase (and mTORC1) activation. Lys may mimick Arg in being sensed by SLC38A9 [109]. With respect to Leu, this AA is known as one of the most potent AAs to activate mTORC1. This effect is mediated by the adaptor protein sestrin2 [110]. Upon Leu binding, sestrin2 dissociates from the GATOR2 complex, a positive regulator of mTORC1 [111]. Subsequently, mTORC1 binds to the cytoplasmic v-ATPase V_1_ subcomplex [61]. This then allows V_1_ to start the re-assembly process with V_0_, so to restore full v-ATPase activity. Arguably, the above-mentioned structural change within V_0_, which is dependent on Arg-induced SCL38A9 activation, may be required to finalize the re-assembly process.

With respect to the opposing AA actions at different concentrations on v-ATPase activation, we offer the following explanation: under basal (AA-deprived) conditions, autophagic proteolysis is expected to be fully active. We speculate that the consequent high intralysosomal AA levels would support relatively high v-ATPase activity via the above-mentioned lysosome-centric inside-out AA sensing mechanism. Subsequently, the addition of low AA concentrations to cells would sufficiently inhibit autophagy and production of intralysosomal AA and thereby prohibit this inside-out signaling, while at the same time, these low concentrations are not sufficient to activate mTORC1 via cytoplasmic sestrin2-mediated AA sensing. Hence, this would elicit a net inhibition of v-ATPase activity. The addition of higher AA concentrations would sufficiently increase this cytoplasmic AA sensing. Furthermore, the increase in cytoplasmic AA would lead to an increase in lysosomal AA (partly through weak base trapping). This would counteract a further inhibition of autophagic proteolysis due to higher cytoplasmic AA concentrations, so that there is a net accumulation of lysosomal AAs, finally allowing SLC38A9-mediated inside-out sensing to additionally contribute to v-ATPase stimulation.

AA-induced assembly of v-ATPase perhaps would be not very effective in (cardiac) cells under (healthy) basal conditions because of relatively high v-ATPase activity (see previous paragraph). Yet under conditions of v-ATPase disassembly as seen in lipid-overexposed cardiac cells, AA (and especially Arg/Leu/Lys) addition could be much more effective. As detailed in Section 3, lipid overexposed cardiac cells display decreased endosomal acidification as a result of lipid-induced v-ATPase disassembly, resulting in CD36 translocation, intracellular lipid accumulation, insulin resistance and contractile dysfunction.

The AA-induced reacidification of endosomes is at the start of a beneficial series of events. First, the reacidified endosomes can again serve as storage compartments for CD36, enabling a net reinternalization of CD36 from the cell surface, thereby reducing myocellular lipid uptake and accumulation. The resulting intracellular lowering of lipid metabolites relieves the block on insulin signaling and insulin-stimulated substrate uptake. Ultimately, the AA (Arg/Leu/Lys) treatment leads to the preservation and/or restoration of contractile function as observed in vivo in rats [61] (Figure 6).

In lipid-overexposed cardiomyocytes, (1a,b) re-addition of AA mixture (arginine/leucine/lysine) results in mTORC1 activation, which attracts the V_1_ subunit to the endosomal membrane, allowing V_1_ to reassemble with V_0_; (2) re-assembly of v-ATPase leads to endosomal acidification; (3a–c) endosomal acidification triggers CD36 retention to the endosome as well as Glut4 translocation to the sarcolemma; (3d) the inhibition of CD36 translocation decreases long-chain fatty acid (FA) uptake, thereby inhibiting lipid accumulation; (4a–c) decreased lipid droplet numbers leads to GLUT4 translocation to the GLUT4 storage vesicle (GSV), which becomes available for insulin-stimulated GLUT4 translocation and insulin-stimulated glucose uptake, therefore shifting energy substrate preference from FA to glucose. Eventually, the re-balancing of energy substrate utilization restores contractile function in lipid-overloaded cardiomyocytes.

## 6. Conclusions and Future Perspectives

In this review, we brought together that lipids, glucose, and AAs impact on v-ATPase assembly status to modulate its proton pumping activity which in turn determines the rate of uptake of these same energy substrates, at least as pertaining to lipids and glucose. However, each substrate employs a different mechanism to regulate v-ATPase. The mechanism by which lipids impact on v-ATPase remains elusive, but for glucose (FBP/aldolase) and AAs (ragulator), the underlying mechanisms have, at least partly, been revealed. In conclusion, v-ATPase thus works as an integrator of nutritional information.

According to the data obtained, v-ATPase assembly by both increased glucose availability and AAs addition might form the basis of novel therapies to combat lipid-induced cardiomyopathy. However, v-ATPase is a large protein complex composed of >14 subunits, several of which exist as more than one isoform, which at present hampers our understanding of the regulatory mechanisms in v-ATPase function. In addition, endosomal acidification not only depends on v-ATPase-mediated proton pumping, but also on the restriction of proton leakage. Although such proton leak in endosomes could be demonstrated upon pharmacological inhibition of v-ATPase [112], little is known about the identity of the proteins responsible for this leak. Furthermore, still emerging is the importance of the v-ATPase in controlling the activity of various signaling pathways, including Wnt, Notch, mTOR, and AMPK signaling [21,113]. As we come to better understand the mechanisms of regulating v-ATPase assembly and trafficking, new possibilities for therapeutic intervention in these signaling pathways arise. Moreover, in addition to the heart the potential roles of v-ATPase re-assembly in lipid-induced insulin resistance and contractile dysfunction also should be investigated in other tissues (e.g., brain, skeletal muscle, and liver). Skeletal muscle and heart show a simliar mechanism of CD36-mediated lipid metabolism. In this respect, further work is still needed to verify whether v-ATPase trafficking, assembly, and function in the control of lipid metabolism operates similarly in skeletal muscle and other tissues.

More specifically, elucidating how amino acids affect v-ATPase is an important step in understanding the basic processes that contribute to cellular homeostasis and the balance of cellular growth that is disrupted in disease states like diabetes [58]. For future perspectives, further studies on the AA-induced m-TORC1–v-ATPase action, which involves a great number of other proteins, including Rag-GTPases, adaptor proteins and lysosomal AA transporters (for review see [105,114], may reveal that several of these other proteins may offer additional, even better druggable, targets to combat lipid-induced cardiomyopathy.

On the other hand, forced v-ATPase disassembly might be an attractive strategy to rectify the substrate balance in heart and muscle metabolic diseases characterized by dyslipidemia [115,116]. The resulting CD36 translocation and an associated increase in fatty acid uptake might restore the substrate balance under these conditions. Not only dyslipidemic conditions might benefit from such strategy, but also metabolic conditions in the heart characterized by excess glucose uptake, such as seen in the pressure overloaded hypertrophic heart [117]. As proof of principle, we have shown that pharmacologically forced CD36 translocation in phenylephrine-induced cardiomyocytes rescues contractile activity in this cell model of cardiac hypertrophy [117].

Finally, as already mentioned, v-ATPase is emerging as a central subcellular hub integrating nutritional information, including at least glucose, palmitate, and AAs. It would be of interest to test the potential effects also of other energy substrates, such as lactate and ketone bodies, on v-ATPase function. This applies especially to the lipid-overloaded diabetic heart, in which v-ATPase activity is impaired. Indeed, there is a great need for detailed studies in diabetic cardiomyopathy in the clinical setting to extrapolate the current findings in experimental animals, isolated heart, and isolated cardiomyocytes to the diabetic patient. The collected data may lead to a better understanding of the processes underlying diabetes and diabetic cardiomyopathy, eventually to novel strategies for treating patients suffering from these diseases.

## Figures and Tables

**Figure 1 metabolites-12-00579-f001:**
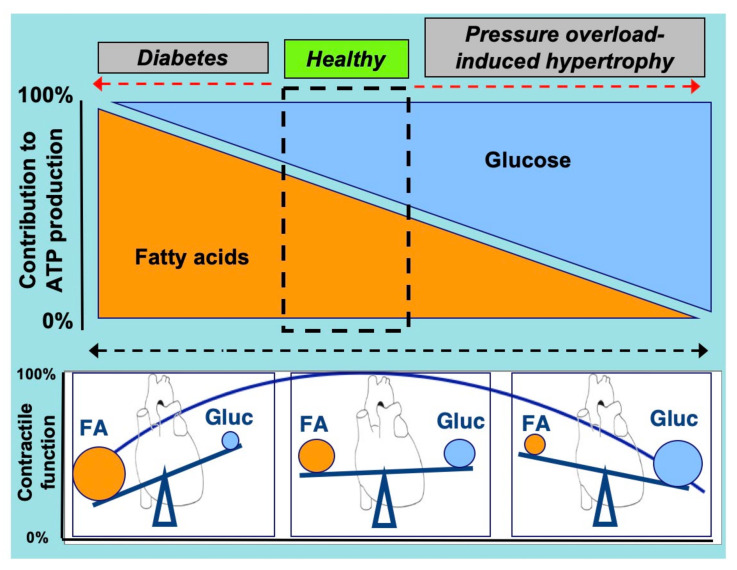
Schematic presentation of the relation between cardiac contractile function and the relative contributions of (long-chain) fatty acids and glucose to overall myocardial ATP production. Not shown are minor contributions from other substrates, such as lactate, ketone bodies, and amino acids. Under healthy conditions, the heart operates optimally when it uses a mixture of energy-providing substrates, especially with respect to fatty acids and glucose. However, if the balance of substrates is tilted, either towards a predominant use of fatty acids (at the expense of glucose) or a predominant use of glucose (at the expense of fatty acids), this change in substrate preference is associated with impaired cardiac contractile function. FA, (long-chain) fatty acids; Gluc, glucose. Adapted from [2].

**Figure 2 metabolites-12-00579-f002:**
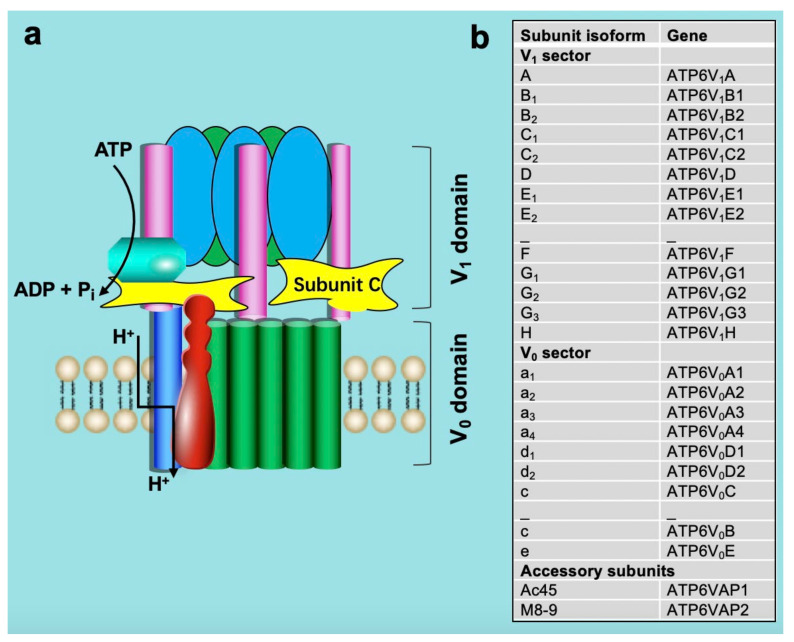
Structure (**a**) and subunit isoform (**b**) of vacuolar-type H^+^-ATPase. See text for detailed explanation.

**Figure 3 metabolites-12-00579-f003:**
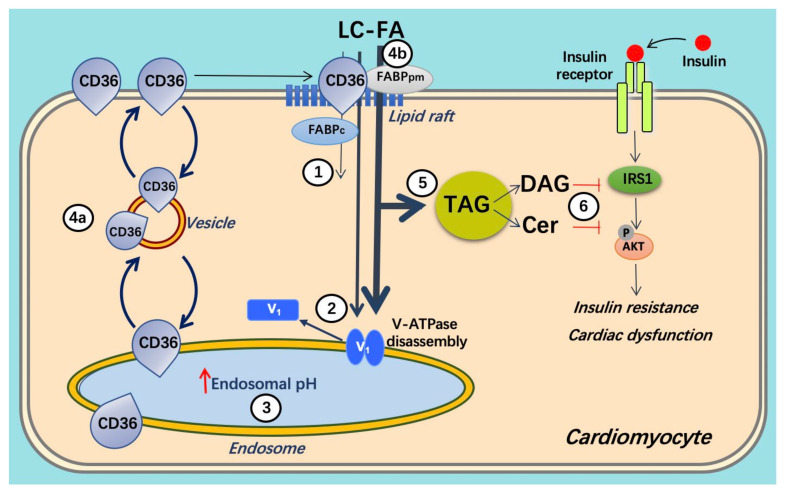
Schematic presentation of lipid oversupply-induced v-ATPase inhibition and its consequences for insulin sensitivity and contractile function. When long-chain fatty acid (FA, e.g., palmitate) supply is high, CD36 translocation from the endosome to the sarcolemma is stimulated. Furthermore, the v-ATPase V_0_ sub-complex, which is integral to the endosomal membrane, is disassembled from the cytosolic V_1_ sub-complex contributing to endosomal alkalinization (or increased endosomal pH). In this situation, the available FA is metabolized to meet the immediate energy demand. However, elevated extracellular FA supply triggers a series of events: (1) increased CD36-mediated FA uptake results in elevated intramyocellular FA levels, (2) FA causes the V_1_ and V_0_ subcomplexes to dissociate, (3) therefore, V_1_ is shifted toward the cytoplasm. (4) The disassembly of v-ATPase leads to endosomal alkalinization. Increased endosomal pH triggers the translocation of CD36 vesicles to the sarcolemma. Upon chronic lipid oversupply, where FA uptake surpasses the metabolic needs, further processes are set in motion. (5) The lipid-induced increase in sarcolemmal CD36 feeds forward to further increased FA uptake and progressive lipid storage. (6) Lipid overload culminates in the loss of insulin sensitivity and contractile function. Adapted from [59].

**Figure 4 metabolites-12-00579-f004:**
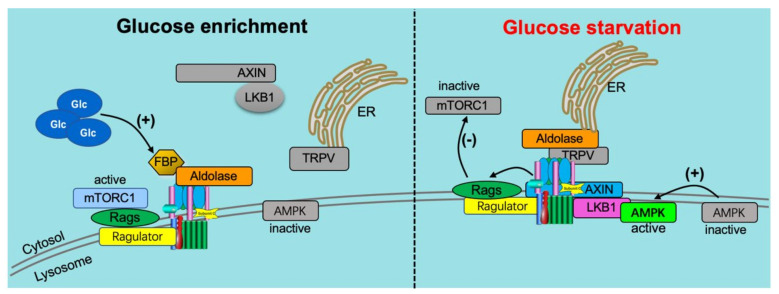
v-ATPase-dependent glucose signaling. (1) When glucose is enriched (**left panel**), it activates the ragulator complex, which acts as a GEF toward the RagGTPases (Rags), activating them. The Rag GTPases are anchored to the lysosomal membrane by the ragulator complex bound to the v-ATPase. Subsequently, the mammalian target of rapamycin complex 1 (mTORC1) is recruited to the lysosomal membrane by the active Rag GTPases, where it can be activated by the lysosomal Rheb GTPases. In addition, when glucose is enriched, the glycolytic intermediate fructose-1,6-bisphosphate (FBP) binds to the aldolase-v-ATPase complex. (2) Under conditions of glucose starvation (**right panel**), FBP disassociates from aldolase, allowing endoplasmic reticulum (ER)-localized transient receptor potential V-type (TRPV) channels to bind the v-ATPase at the aldolase binding site. FBP-unoccupied aldolase binds and inhibits TRPV Ca2þ channel activity allowing for the recruitment of the AXIN-LKB1 complex to the lysosomal membrane where it interacts with ragulator and inhibits its GEF activity toward Rags causing the disassociation from the lysosome and inactivation of mTORC1. Furthermore, the v-ATPase, AXIN, and LKB1 recruit AMPK to form a super complex where AMPK is phosphorylated and activated by LKB1. Adapted from [21].

**Figure 5 metabolites-12-00579-f005:**
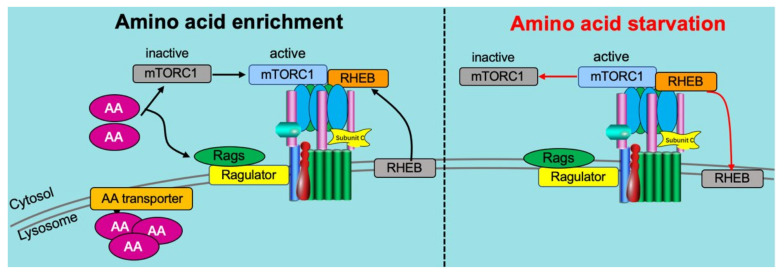
v-ATPase-dependent amino acid signaling. (1) Under conditions of high amino acid (AA) availability (**left panel**), intralysosomal AA activates the ragulator complex, which acts as a GEF toward the Rag GTPases (Rags), activating them. The Rag GTPases are anchored to the lysosomal membrane by the Ragulator complex bound to the V-ATPase. Subsequently, the inactive mammalian target of rapamycin complex 1 (mTORC1) is recruited to the lysosomal membrane by the active Rag GTPases, where it can be activated by the lysosomal Rheb GTPases. (2) Under low AA conditions (**right panel**), Ragulator GEF activity is inhibited, inactivating Rags, and leading to the disassociation from the lysosome and inactivation of mTORC1. Adapted from [21].

**Figure 6 metabolites-12-00579-f006:**
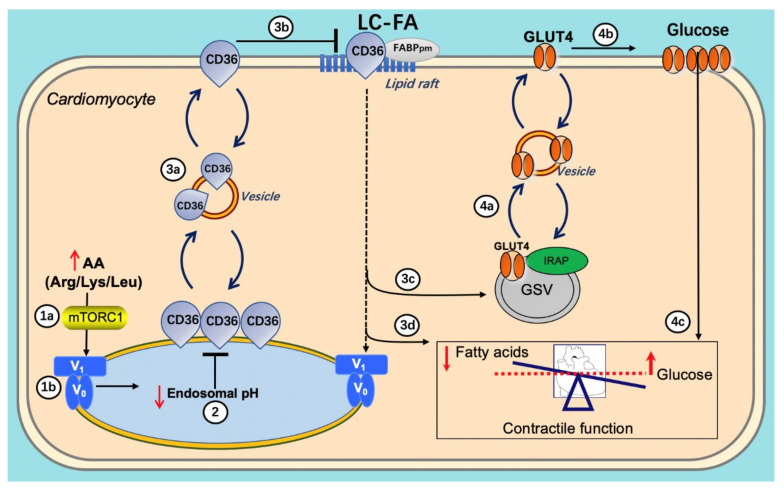
Schematic presentation of the re-assembly of v-ATPase as a target to restore contractile function in lipid-overloaded heart cells.

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
