# Peer review of "Endosomal v-ATPase as a Sensor Determining Myocardial Substrate Preference"

_metabolites, 2022, doi:10.3390/metabo12070579_

Round 1

Reviewer 1 Report

This effect is well written review paper.

The reviewer only has one minor comments.

Authors discussed the link between Sestrin2 and mTORC1 activity in section 5.2. Sestrin2 is a component of AMPK and LKB1.  Genetic elimination of sestrin2 affects AMPK activation. Could authors include sestrin2 in figure 4 and discuss the potential role in AMPK activation in section 4.1.

Author Response

Reviewer 1:

We thank this reviewer of having critically evaluated our manuscript and for her/his praising words. Below, please find the answer to the minor comment.

Authors discussed the link between Sestrin2 and mTORC1 activity in section 5.2. Sestrin2 is a component of AMPK and LKB1.  Genetic elimination of sestrin2 affects AMPK activation. Could authors include sestrin2 in figure 4 and discuss the potential role in AMPK activation in section 4.1.

Reply:

Thanks for suggesting this interesting link. We added the following information (to section: “Further, it has been reported that Sestrin2, a metabolic regulator that responds to various stresses including nutrition, can cause AMPK activation (reference 81), in line with possible involvement of Sestrin2 in regulation of V-ATPase assembly status (as further discussed in section 5.2).”

Reviewer 2 Report

The review authored by Wang et. al. addresses some cellular aspects involving the impact of fatty acids, glucose and AAs on the assembly of the vacuolar-type H+-ATPase, particular in the pathological context of failing heart. I find that the manuscript is well written and could be considered for publication after some minor modifications.

Major:

Structural aspects: A sequence alignment with ATP-ase sequence between different species should be added to delineate protein conservation across taxa.

Minor: 

1.     P7: “…which appears to involve v- ATPase, In particular, v-ATPase activity…” probably should be “…which appears to involve v- ATPase. In particular, v-ATPase activity…”

2.     Figure 4 legend: “When glucose is enrichment…” should probably be “When glucose is enriched…”.

Author Response

Reviewer 2:

We thank this reviewer of having critically evaluated our manuscript and for her/his praising words. Below, please find the answer to the comments in a point-by-point manner.

Major:

Structural aspects: A sequence alignment with ATP-ase sequence between different species should be added to delineate protein conservation across taxa.

Reply:

A sequence alignment of v-ATPase between different species would be a bit out of topic in view that this review focusses on the involvement of v-ATPase in regulation of metabolism only in the mammalian heart. In addition, this would have been an immense task, given that v-ATPase consists of 14 subunits. Nonetheless, we have now provided information that the subunit composition and regulation by assembly appears to be conserved from yeast to mammals, also implying that the role of v-ATPase in regulating metabolism might be conserved. For this, see two new textual additions in this section highlighted in yellow (page 6 and page 8 of revised manuscript).

Minor:

  1. P7: “…which appears to involve v- ATPase, In particular, v-ATPase activity…” probably should be “…which appears to involve v- ATPase. In particular, v-ATPase activity…”

Reply:

Thanks for detecting this error. We now have made the suggested change (see yellow highlighted text).

  1. Figure 4 legend: “When glucose is enrichment…” should probably be “When glucose is enriched…”.

Reply:

Thanks for detecting this error. We now have made the suggested change.

Reviewer 3 Report

Wang et. al. presented a very interesting review about role of v-ATPase in regulation of metabolic substrate flux (mainly to cardiac cells). Although the article is well written and covers main topics, there are small issues, the authors need to consider.

1. In the introduction section authors could extend information about resarch indicated the competition beetwen glucose and FFA (after citation 6). Valuable will we studies focused on NMR analysis or novel methods investigated substate preference for example : "Indexing Tricarboxylic Acid Cycle Flux in Intact Hearts by Carbon-13 Nuclear Magnetic Resonance". Circ. Res., 1992, 70 (2). https://doi.org/10.1161/01.RES.70.2.392.; Jeffrey, F. M. H.; Roach, J. S.; Storey, C. J.; Sherry, A. D.; Malloy, C. R. 13C Isotopomer Analysis of Glutamate by Tandem Mass Spectrometry. Anal. Biochem., 2002, 300 (2). https://doi.org/10.1006/abio.2001.5457. ; Comprehensive Quantification of Fuel Use by the Failing and Nonfailing Human Heart. Science (80-. )., 2020, 370 (6514). https://doi.org/10.1126/science.abc8861. or igh Throughput Procedure for Comparative Analysis of In Vivo Cardiac Glucose or Amino Acids Use in Cardiovascular Pathologies and Pharmacological Treatments. Metabolites. 2021; 11(8):497. https://doi.org/10.3390/metabo11080497".

2. In the part "v-ATPase and CD36 traffic", page 6, last paragraph authors should consider that are a specific cases in wich high FFA levels and metabolic swich induce a possitive metabolic and functional effect on heart or skeletal muscle ref.  Enhanced Cardiac Hypoxic Injury in Atherogenic Dyslipidaemia Results from Alterations in the Energy Metabolism Pattern. Metabolism., 2021, 114. https://doi.org/10.1016/j.metabol.2020.154400.;  Enhanced Muscle Strength in Dyslipidemic Mice and Its Relation to Increased Capacity for Fatty Acid Oxidation. International Journal of Molecular Sciences. 2021; 22(22):12251. https://doi.org/10.3390/ijms222212251.

3. In the part " Conclusion and future perspectives" authors should add more general information about the main directions of theraphies focused on glucose, FFA or AA shifts in heart, its reduction or increasement will be valuble, in which patological conditions.

Author Response

Reviewer 3:

We thank this reviewer of having critically evaluated our manuscript and for her/his praising words. Below, please find the answer to the comments in a point-by-point manner.

Comments:

  1. In the introduction section authors could extend information about resarch indicated the competition beetwen glucose and FFA (after citation 6). Valuable will we studies focused on NMR analysis or novel methods investigated substate preference for example : "Indexing Tricarboxylic Acid Cycle Flux in Intact Hearts by Carbon-13 Nuclear Magnetic Resonance". Circ. Res., 1992, 70 (2). https://doi.org/10.1161/01.RES.70.2.392.; Jeffrey, F. M. H.; Roach, J. S.; Storey, C. J.; Sherry, A. D.; Malloy, C. R. 13C Isotopomer Analysis of Glutamate by Tandem Mass Spectrometry. Anal. Biochem., 2002, 300 (2). https://doi.org/10.1006/abio.2001.5457. ; Comprehensive Quantification of Fuel Use by the Failing and Nonfailing Human Heart. Science (80-. )., 2020, 370 (6514). https://doi.org/10.1126/science.abc8861. or igh Throughput Procedure for Comparative Analysis of In Vivo Cardiac Glucose or Amino Acids Use in Cardiovascular Pathologies and Pharmacological Treatments. Metabolites. 2021; 11(8):497. https://doi.org/10.3390/metabo11080497".

Reply:

Thank you very much for alerting us to these four publications. We have now incorporated the last two refs in our reference list (the first two refs are appreciated, but not so novel anymore). Please note that the second last reference stated in the abstract that “both heart and leg consumed ketones, glutamate, and acetate in direct proportionality to circulating levels, indicating that availability is a key driver for consumption of these substrates”. This is very much in line with our concluding sentence in this section that “at any specific time, however, the actual contribution of each of these substrates (ketones and AAs) could be quite different and be adjusted rapidly, depending on changes in substrate availability governed by specific (patho)physiologic conditions”.

  1. In the part "v-ATPase and CD36 traffic", page 6, last paragraph authors should consider that are a specific cases in wich high FFA levels and metabolic swich induce a possitive metabolic and functional effect on heart or skeletal muscle ref. Enhanced Cardiac Hypoxic Injury in Atherogenic Dyslipidaemia Results from Alterations in the Energy Metabolism Pattern. Metabolism., 2021, 114. https://doi.org/10.1016/j.metabol.2020.154400.; Enhanced Muscle Strength in Dyslipidemic Mice and Its Relation to Increased Capacity for Fatty Acid Oxidation. International Journal of Molecular Sciences. 2021; 22(22):12251. https://doi.org/10.3390/ijms222212251.

Reply:

Indeed this reviewer is entirely correct that under specific metabolic conditions a forced increase in fatty acid uptake could be beneficial. Yet, our review was focused on diabetic cardiomyopathy characterized by v-ATPase disassembly and lipid-induced insulin resistance. Notwithstanding it is a valuable point to make that forced disassembly of v-ATPase could be beneficial under conditions of dyslipidemia. We now have dedicated a specific paragraph in the Discussion (new second last paragraph, highlighted in yellow, page 24 of revised manuscript) to emphasize this issue. We also included both references.

  1. In the part " Conclusion and future perspectives" authors should add more general information about the main directions of theraphies focused on glucose, FFA or AA shifts in heart, its reduction or increasement will be valuble, in which patological conditions.

Reply:

In line and continuation with the previous comment, we have further extended this new paragraph to state more generally that forced CD36 translocation can be used as strategy to combat not only cardiac dyslipidemia but also cardiac metabolic diseases characterized by excess glucose uptake (see text highlighted in green).